# Correlating Ocular Physiology and Visual Function with Mild Cognitive Loss in Senior Citizens in Taiwan

**DOI:** 10.3390/jcm11092624

**Published:** 2022-05-06

**Authors:** Kuo-Chen Su, Hong-Ming Cheng, Yu Chu, Fang-Chun Lu, Lung-Hui Tsai, Ching-Ying Cheng

**Affiliations:** 1Department of Ophthalmology, Chung Shan Medical University Hospital, Taichung 402, Taiwan; jimmysu8@csmu.edu.tw; 2Department of Optometry, Chung Shan Medical University, Taichung 402, Taiwan; tom830107@gmail.com (Y.C.); lu700112@gmail.com (F.-C.L.); 3Department of Optometry, Asia University, Taichung 413, Taiwan; hm_cheng@yahoo.com

**Keywords:** cognitive function, binocular vision, mild cognitive impairment, ocular physiology, dementia

## Abstract

**Purpose:** The transition of Taiwan from an aging to a super-aging society has come with a cost as more elderly now suffer from cognitive impairment. The main purpose of our study was to investigate if early detection can be developed so that timely intervention can be instituted. We analyzed the correlation of cognitive function with ocular physiology and visual functions between senior citizens aged 60 years or older in Taiwan. **Methods:** Thirty-six healthy subjects were recruited for the study. Addenbrooke’s cognitive examination III (ACE-III), binocular functions (including objective and subjective refraction, distance and near dissociated phoria, stereopsis, contrast sensitivity, adult developmental eye movement (ADEM), and ocular physiology (by using optical coherence tomography, OCT, and macular pigment measurement, MPS) were performed, and the data were analyzed via independent *t*-test, chi-square test, Pearson correlation, linear regression, and ROC (receiver operating characteristic) curve. **Results:** Data analysis showed that (1) patients with poor eye movement had a strong correlation with the total score and all dimensions of cognitive functions, (2) the thickness of the macula had a strong correlation with attention and memory, and (3) patients with poor eye movement and poor stereopsis in combination with thinner inferior macula appeared to have lower cognitive abilities. **Discussion and Conclusions**: Cognitive dysfunction is not readily identified during the early stage of cognitive decline. The use of simple and inexpensive ADEM or stereopsis test and comparing the OCT results that are popular in optometry clinics for reference can be diagnostic in identifying patients with mild cognitive impairments. With the combined use of macular pigment density or retinal thickness measurements, it was possible to effectively predict the early degradation of cognition.

## 1. Introduction

Alzheimer’s disease is the most common type of dementia [1,2,3]. Studies have shown that many patients with Alzheimer’s disease (AD) have reading difficulties resulting from line skipping, mainly because of degeneration in fixation, saccadic pursuit, or vergence capabilities [4,5]. Although such changes are usually subtle in the early stages of dementia, or mild cognitive impairment (MCI), they can still be detected through a functional vision assessment [5]. In terms of saccades, Pirozzolo and Hansch compared 12 Alzheimer’s patients with a control group, and found that the saccadic efficiency of Alzheimer’s patients increased significantly [4], which suggests a change in higher cortical regulatory roles in sensory-motor integration. Although Alzheimer’s patients have normal amplitudes in visual evoked potential (VEP) examination, the change in latency shows that the patients have difficulty interpreting visual information to some extent [6]. Additional studies have also demonstrated that contrast sensitivity, pupil response, color vision, fixation, saccadic eye movement, etc. are implicated in functional deficits in the early stages of dementia [7,8,9,10]. Furthermore, while binocular vision problems are already common in the elderly population [11], relevant studies have pointed out that the visual function of the elderly with MCI is even worse [12].

In terms of visual physiology, retinal examination can provide a non-invasive method which is similar to the examination of brain pathology [13]; previous literature has pointed out that the reduction in macular thickness [14,15], thinning of retinal nerve fiber layers, changes in the optic nerve or optic disc [16,17], and macular pigment density [18] are closely related to dementia progression. Kim and Kang’s 2019 study [19] showed that the macular ganglion intracellular plexiform layer (GC-IPL) and total macular and peripheral retinal nerve fiber layer (RNFL) in the Alzheimer’s group were significantly thinner than those in the control group [19]; Claire and Michèle [20] also indicated that, when comparing with the control group, the thickness of RNFL in patients with MCI, mild AD, or moderate to severe AD was significantly decreased.

To summarize, the degradation of cognition may have a causal relationship with the overall binocular visual functions and ocular physiology that is yet to be explicitly explored. The purpose of our study was to demonstrate, for the first time, that the decline in visual function and ocular physiology may be correlated and even interact significantly with MCI.

## 2. Materials and Methods

The study was a cross-sectional study that was conducted from 20 November 2020 to 30 March 2021 at Chung Shan Medical University Hospital (CSMUH)-affiliated Dementia Intergraded Care Center. All the procedures were conducted in accordance with the Declaration of Helsinki. Approval was obtained from the Institutional Review Board of CSMUH (Taichung, Taiwan) (approval number: CS19110). The age and physical, spirit, and compliance situations may have determined the duration for each subject to complete the entire examination. The manuscript is reported according to the STROBE guideline [21,22].

### 2.1. Research Subjects

All participants had binocular and monocular distance and near visual acuity of 0.8 or better, spherical power ranged from −5.00 D to +2.00 D, and astigmatism was <1.00 D. Twenty-three subjects were already excluded at the recruiting stage owing to eye and/or mental problems. Those with an optic disc ratio not within the normal range could be a high-risk group with high myopia or glaucoma, and were also excluded and referred. A total of 40 people eventually participated and four were later dismissed, one with poor communication because of auditory nerve damage, and three others simply dropped out. Finally, the effective sample was 36 with 6 men and 30 women, and the average age was for men: 68.83 ± 9.07, for women: 73.67 ± 9.44 years, and for all subjects was 72.86 ± 9.43 years. An independent *t*-test indicated no significant difference in age and ACE-III cognitive score as far as sex; therefore, the results were analyzed with all 36 subjects combined.

### 2.2. Research Materials

Addenbrooke’s cognitive examination III (ACE-III, Chinese version) was performed for differentiating patients with or without cognitive impairment. The sensitive of ACE-III is reported to be trustworthy for detecting early stages of dementia, and is available in different languages [23]. ACE-III contains 5 sub-tests with a total of 100 questions, including attention (18 questions), memory (26 questions), language fluency (14 questions), language comprehension (26 questions), and visuospatial ability (16 questions). The criteria of ACE-III is 83 [23], i.e., subjects who finish with 83 points or less are regarded as being ACE-abnormal.

Binocular visual function examination included subjective refraction (Shin-Nippon Wide-View Refraction NVision-K 5001, Tokyo, Japan), distance and near visual acuity (View-M digital visual acuity chart and TMVC near point test card), phoria (Howell card for distance and near), adult developmental eye movement test (ADEM), Stereo-Fly for stereo acuity test, and the low contrast flip chart for near and distance contrast sensitivity. In order to avoid potential bias, each test was performed by the same optometrist. ADEM was developed by Gené-Sampedro et al. [24] which was modified from the DEM test [25], and although DEM has been suggested to be potentially useful for adults [26], the norm of DEM was only constructed up to the age of 14, and modification was therefore necessary [27,28]. Subjects were asked to read a series of numbers as fast as possible which were arranged either horizontally or vertically. Four possible behavioral types can be found with the ADEM test: normal, ocular movement dysfunction, naming disorder, and combined.

Ocular physiology examination included optical coherence tomography (OCT, Topcon, MAESTRO2, Tokyo, Japan) and macular pigment measurement (Macular Pigment Screener II, MPS-II), both of which were performed on patients without mydriasis, in a dim light room. In the examination of macular thickness with OCT within the center area (recorded as a circle with a diameter of 1 mm), the inner area (a circle with a diameter of 1 mm to 3 mm), and the outer area (a circle with a diameter of 3 mm to 6 mm). With OCT, the retinal thickness of the inner and outer area was divided into four quadrants: superior, inferior, nasal, and temporal. Macular pigment is believed to play a beneficial role in visual performance [29,30]; therefore, subjects were asked to perform three times and were first instructed to push a button as soon as a flicker was perceived.

### 2.3. Data Analysis and Statistical Analysis

G* Power analysis was used to determine the sample size in this study, in which the condition was set at the effect size d = 0.8, α = 0.05, power (1 − β) = 0.95. Data were analyzed by using SPSS 26.0 statistical software (IBM, Armonk, NY, USA). For the statistics including (1) independent *t*-test for comparing binocular visual functions and ocular physiology between ACE groups; (2) chi-square analysis between the ACE group and ADEM types, distance, and near phoria; (3) linear regression analysis about cognitive function, visual function, and ocular physiology; (4) ROC curve between significant variables; and (5) Pearson correlation analysis between OCT, ADEM, MPS, and stereopsis, a value of *p* < 0.05 was considered statistically significant.

## 3. Results

The ACE-III test contains attention, memory, language fluency, language comprehension, and visuospatial categories, and the reliability of the ACE-III cognitive examination was excellent with Cronbach’s alpha coefficient = 0.82. In addition, 15 ACE-normal and 21 ACE-abnormal subjects were found by the ACE score. G* Power analysis showed that there was still enough power after recalculation and adjustment.

The results of the independent *t*-test showed that cognitive function showed significant differences not only in the total score of ACE (*t* = −8.93, *p* = 0.00) but also in all subtests: attention (*t* = −4.54, *p* = 0.00), memory (*t* = −9.89, *p* = 0.00), language fluency (*t* = −5.06, *p* = 0.00), language comprehension (*t* = −5.26, *p* = 0.00), and visuospatial (*t* = −6.58, *p* = 0.00), indicating that the degradation of cognitive function had a wide range of effects (Table 1).

### 3.1. Ocular Physiology between ACE-Abnormal and ACE-Normal Groups

All data were collected after confirming that subjects had no existing eye diseases and risks; at the same time, all participants had eligible refractive errors and visual acuity. In analyzing visual physiology, because the data from the left and right eyes were not significantly different, the data shown here are from the right eye. The density of macular pigment was significantly different (*t* = −2.50, *p* = 0.02) where that in the ACE-abnormal group was lower than that in the ACE-normal group (Table 2).

In the OCT examination of the macula, except for the central area (OCT-Center-Fovea: *t* = −0.65, *p* = 0.52), the inner area of temporal (OCT-Inner-Temporal: *t* = −1.83, *p* = 0.08) and nasal (OCT-Inner-Nasal: *t* = −1.67, *p* = 0.10) quadrants were not significantly different, but the rest of the macular thickness was significantly different between the two groups: there were obvious differences in the peripheral part of the macula, especially thickness of the outer, the inferior, and the superior areas (Table 2, Figure 1).

### 3.2. Visual Functions between ACE-Abnormal and ACE-Normal Groups

In term of visual functions, eye movement (ADEM), stereopsis, contrast sensitivity, distance phoria, and near phoria were performed. The difference and the analysis results with the independent sample *t*-test are listed in Table 3. The performance of the two groups in ADEM_V was significantly different (*t* = −5.70, *p* = 0.000), and the ADEM_V of the ACE-abnormal group was significantly less than the ACE-normal group; the same results also appeared in ADEM_H (*t* = 6.24, *p* = 0.000), ADEM_Ratio (*t* = 5.63, *p* = 0.000), and stereopsis (*t* = 5.95, *p* = 0.006). Only contrast sensitivity (*t* = 1.45, *p* = 0.16) did not reach a significant difference between the two groups.

#### 3.2.1. ACE Cognitive Function and ADEM Eye Movement

Adult saccade eye movement, including vertical time, horizontal time, and ADEM ratio: in the ACE-abnormal group, 3 patients had the normal type, 2 had difficulty in naming, 16 had combined disorder, and none had only eye movement disorder. In the ACE-normal group, 17 had the normal type of ADEM, and 1 had difficulty in naming (as shown in Figure 2). Chi-square analysis was used to analyze the differences between ADEM types and the cognitive function, and the results showed that eye movement abilities had significant difference (Chi-square *=* 21.96, *p =* 0.000) in cognitive function, i.e., there was a strong correlation between eye movement and cognitive functions. It is worthwhile to mention that the proportion of ADEM combined disorders in the ACE-abnormal group was much higher than the ACE-normal group, which showed that in the ADEM of patients with cognitive dysfunction, the probability of combined naming and eye movement difficulties was significantly higher in the ACE-abnormal group.

#### 3.2.2. ACE Cognitive Function and Phoria

Distant phoria test showed: in the ACE-abnormal group, there were seven patients with orthophoria, five with esophoria, seven with exophoria, and two with monovision; in the ACE-normal group, six patients were with orthophoria, six with esophoria, and three with exophoria. (as shown in Figure 3). Chi-square analysis showed that ACE cognition was not correlated with phoria (Chi-square = 2.847, *p* = 0.416). The near phoria test showed: in the ACE-abnormal group, there were 4 patients with orthophoria, 3 with esotropia, 12 with exophoria, and 2 with monovision; in the ACE-normal group, 3 patients have esophoria and 12 patients with exophoria (as shown in Figure 3). Same as distance phoria, near phoria also showed no significant difference (Chi-square = 4.261, *p* = 0.235) in cognitive function. The results indicated that there was no correlation between eye position and cognitive function [31].

### 3.3. Indicator for Cognitive Function

#### 3.3.1. Linear Regression Analysis on Significant Variables

According to the analysis above, there were significant differences between normal and abnormal cognitive groups in macular pigment, retinal thickness, eye movement ability, and stereopsis. If the ACE total score was taken as the dependent variable, and macular pigment, retinal thickness, eye movement ability, and stereopsis were all taken as independent variables for linear regression analysis, the results showed that ADEM horizonal, stereopsis, and retinal thickness (outer layer, inferior quadrant) had a high ability to predict and explain cognition decline. The three independent variables could be used to explain up to 82.1% (adjusted to 80.4%) of the explained variation generated by the cognitive ability dependent variables (Table 4).

Furthermore, ADEM-V and retinal thickness (outer layer, inferior quadrant, OCT_O_I) also had a high ability to predict and explain ACE attention (71.6%), ADEM-Ratio and OCT_O_I could predict ACE memory for up to 81.0%, ADEM-H and stereopsis could explain 55.5% of ACE language fluency, ADEM-V and stereopsis could explain 68.1% of ACE language comprehension, and ADEM-H could explain 43.8% of ACE visuospatial.

The data showed that horizontal eye movement was related to language fluency and visuospatial abilities; vertical eye movement to attention and language comprehension; the overall eye movement ability to memory; the thickness of the retina (OCT_O_I) was related to attention, memory, and overall cognitive ability; and stereopsis to language fluency, language comprehension, and overall cognitive ability.

#### 3.3.2. ROC Curve Analysis about Significant Variables

Since eye movement, stereopsis, and retinal thickness showed good explanatory capabilities for cognitive ability, the critical criteria (cut-off point) value of various tests in clinical examination was worthy of further discussion. The results of the ROC curve analysis showed that eye movement examination had the most discriminative ability of clinical screening, the AUC (area under the curve) of horizontal eye movement (ADEM_H) was 0.924 (*p* = 0.000), sensitivity = 0.857, and specificity = 0.867, and clinical identification could set the cut-off point at 38.5 s. Similarly, other eye movement examinations (vertical eye movement: AUC = 0.914, *p* = 0.000, cut-off point = 35.5; eye movement ratio: AUC = 0.902, *p* = 0.000, cut-off point = 1.12) could also be used as clinically important reference values.

In addition, the thickness of the retina (OCT_O_I) and stereopsis also showed good to excellent discriminative abilities, with which OCT_O-I performed at AUC = 0.795, *p* = 0.009, cut-off point = 243 mm, and stereopsis performed at AUC = 0.794, *p* = 0.012, the cut-off point = 100. All variables could be cross-validated before referral and identification(Table 5 and Figure 4).

## 4. Discussion

In this study, an ACE score of 83 was used as the cut-off point, and all subjects were divided into either the ACE-normal (>83) or ACE-abnormal (≤83) group. There were significant differences in the performance of the total score and six cognitive dimensions. On the whole, this cognitive screening questionnaire showed great reliability and validity, and it could effectively identify patients with mild cognitive impairments. However, ACE examination was time-consuming, and subjects were required to have sufficient cooperation.

Despite these shortcomings, the biggest difference between the ACE-normal cognitive function group and the abnormal group was the proportion of eye movement disorders. With the ADEM test, for patients with abnormal cognitive function, the probability of combined naming and eye movement difficulty was significantly higher than that in the normal group. ADEM has excellent discriminative power in predicting cognitive states under the linear regression and ROC curve, so ADEM can effectively identify the presence of cognitive impairment [25]; in addition, ADEM is inexpensive, rapid, and simple, and it is an ideal tool for the screening of cognitive impairment.

Our present study further pointed to two possibly simpler and more effective screening methods that can be conducted in a clinical setting:In terms of visual physiology, the macular pigment density of the ACE-abnormal group was significantly lower than that of the ACE-normal group, and there was a significant difference in the peripheral thickness of the macular (the superior, inferior, and temporal of the outer macular layer) between the normal and the abnormal groups. Previous studies have indicated that the superior and inferior layers are the most obvious [32] in patients with cognitive impairment. Our results further indicated the outer and inferior macular layers as having excellent discrimination ability in predicting cognitive states under the linear regression and ROC curve. Fundus examination of patients can therefore be used to effectively detect the developing decline of the cognitive state [33,34]. According to the previous studies and the results of this study, we can assume that the thinning of the ganglion cell layer leads to the disappearance of neurons, thereby affecting the dorsal and the ventral pathways; in addition, these neural changes contribute to age-related losses of low-level visual functions and even higher-order visual perceptions, including face perception [35], motion processing [36,37,38], and reading speed [39].Moreover, linear regression indicated that ADEM, stereopsis, and macular thinness interact to some extent. For example, ADEM-V, ADEM-H, and stereopsis appeared in the cognitive function of language comprehension and language fluency, presumably because the ADEM measurement can be classified as a group of naming disorders or a combined disorder [25]. When it comes to stereopsis, the reason it is correlated with language comprehension and language fluency might be traced back to the pyramid of binocular vision development [40,41].

We should point out that procedures (1) and (2) described above, for monitoring cognitive impairment through readily available testing of eye movement and stereopsis, and OCT measurement of retinal thickness, must be performed as a whole. In a well-equipped eye clinic, this is not a problem. Otherwise, the correlation between single tests and cognitive function may not be as high as desired (see Figure 5, the correlation coefficient was between 0.40 and 0.47), and referrals for additional tests may become necessary.

## 5. Conclusions

Mild cognitive impairment (MCI) is a sign and also a necessary process before entering dementia; in the initial stage, it is often considered to be a due phenomenon of aging and ignored. There appears concurrent drastic changes in ocular physiology and visual function. In fact, eye movement, stereo vision, macular thinness are all indicators in the beginning stage of dementia. A reliable testing for MCI should also be a non-investment, more convenient, low-cost, and less time-consuming. Our results showed that testing of visual function and ocular physiology was suitable for the rapid screening of patients with MCI before the onset of overt mobile and cognitive disabilities.

## Figures and Tables

**Figure 1 jcm-11-02624-f001:**
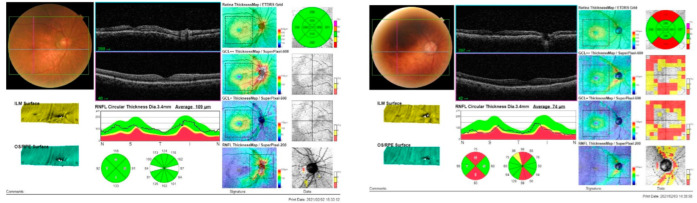
OCT measurement: examples of one ACE-normal case (**left**) and an ACE-abnormal case (**right**).

**Figure 2 jcm-11-02624-f002:**
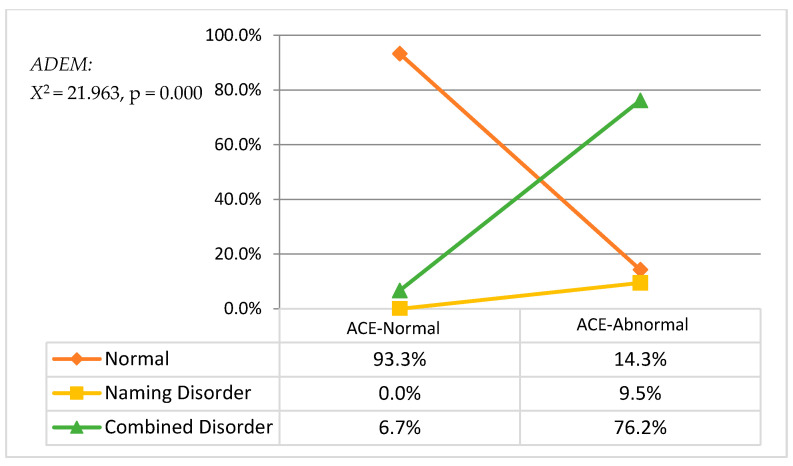
ADEM types between ACE-normal and -abnormal subjects.

**Figure 3 jcm-11-02624-f003:**
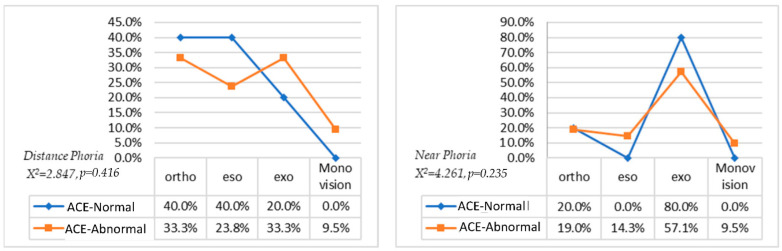
Distance phoria and near phoria between ACE-normal and -abnormal subjects.

**Figure 4 jcm-11-02624-f004:**
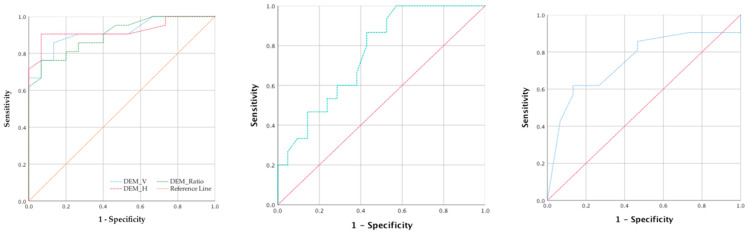
ROC curve analysis about ADEM, OCT_O_I, and stereopsis.

**Figure 5 jcm-11-02624-f005:**
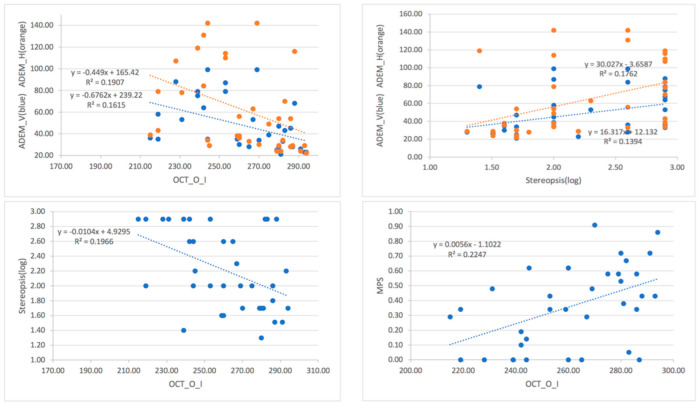
Correlations between OCT_O_I, MPS, Stereopsis, and ADEM values.

**Table 1 jcm-11-02624-t001:** Independent *t*-test of ACE-III total score and subtests between ACE groups.

	ACE-Abnormal N = 21/M (SD)	ACE-Normal N = 15/M (SD)	Levene *F* Value	*t*	*p*
ACE Total score	63.00 (12.64)	88.93 (3.51)	19.24	−8.93	0.000 **
ACE Attention	13.57 (3.71)	17.40 (0.91)	19.21	−4.54	0.000 **
ACE Memory	14.05 (2.78)	22.60 (2.20)	0.07	−9.89	0.000 **
ACE Language fluency	5.14 (2.73)	9.33 (1.99)	2.66	−5.06	0.000 **
ACE Language comprehension	19.95 (4.40)	25.13 (0.83)	20.53	−5.27	0.000 **
ACE Visuospatial	10.29 (2.24)	14.47 (1.19)	4.03	−6.58	0.000 **

** *p* < 0.01.

**Table 2 jcm-11-02624-t002:** Independent *t*-test of ocular physiology between ACE-normal and -abnormal subjects.

	ACE-Abnormal N = 21/M (SD)	ACE-Normal N = 15/M (SD)	Levene *F* Value	*t*	*p*
MPS	0.27 (0.22)	0.49(0.30)	1.17	−2.50	0.02 *
OCT-Center-Fovea	250.05 (35.55)	257.07(26.01)	0.11	−0.65	0.52
OCT-Inner-Temporal	296.52 (27.12)	311.40(18.63)	1.74	−1.83	0.08
OCT-Inner-Superior	305.52 (27.79)	330.00(20.04)	1.23	−2.91	0.01 *
OCT-Inner-Nasal	310.29 (26.84)	324.93(24.54)	0.10	−1.67	0.10
OCT-Inner-Inferior	301.14 (24.63)	320.53(18.30)	0.74	−2.58	0.01 **
OCT-Outer-Temporal	251.00 (32.74)	270.40(13.55)	4.84	−2.44	0.02 *
OCT-Outer-Superior	260.15 (34.32)	293.13(24.52)	1.85	−3.16	0.003 **
OCT-Outer-Nasal	278.38 (26.63)	307.13(16.32)	2.96	−3.70	0.001 **
OCT-Outer-Inferior	252.95 (23.70)	273.07(16.78)	2.73	−2.82	0.008 **

* *p* < 0.05, ** *p* < 0.01.

**Table 3 jcm-11-02624-t003:** Independent *t*-test of visual function between ACE-normal and -abnormal subjects.

	ACE-Abnormal N = 21/M (SD)	ACE-Normal N = 15/M (SD)	Levene *F* Value	*t*	*p*
ADEM_V	60.86 (23.42)	30.20 (6.50)	27.51	5.70	0.000 **
ADEM_H	84.38 (37.41)	31.87 (8.03)	33.45	6.24	0.000 **
ADEM_Ratio	1.35 (0.20)	1.06 (0.12)	5.07	5.63	0.000 **
Stereo	2.40 (0.55)	1.93 (0.41)	4.23	2.95	0.006 **
Contrast sensitivity	1.37 (0.38)	1.25 (0.00)	7.45	1.45	0.16

ADEM: adult developmental eye movement; VT vertical time; and HT: horizontal time. ** *p* < 0.01.

**Table 4 jcm-11-02624-t004:** Linear regression analysis on significant variables.

Dependent Variables		R	R^2^	Adjusted R^2^	SE	Change Value			F	Sig.
						R^2^ Change	F Change	*p* for F Change		
ACE total score	1. ADEM-H	0.841	0.707	0.698	9.03	0.707	79.638	0.000	79.638	0.000 **
	2. 1 + Stereopsis	0.890	0.792	0.779	7.72	0.085	13.158	0.001	61.069	0.000 **
	3. 2 + OCT-O-I	0.906	0.821	0.804	7.29	0.029	4.946	0.034	47.382	0.000 **
ACE attention	1. ADEM-V	0.824	0.679	0.670	2.00	0.679	69.930	0.000	69.930	0.000 **
	2. 1 + OCT-O-I	0.846	0.716	0.699	1.91	0.037	4.178	0.049	40.422	0.000 **
ACE memory	1. ADEM-Ratio	0.825	0.681	0.672	2.85	0.681	70.564	0.000	70.564	0.000 **
	2. 1 + OCT-O-I	0.900	0.810	0.698	2.23	0.129	21.633	0.000	68.159	0.000 **
ACE language fluency	1. ADEM-H	0.674	0.455	0.438	2.35	0.455	27.538	0.000	27.538	0.000 **
	2. 1 + Stereopsis	0.745	0.555	0.528	2.15	0.101	7.238	0.011	19.990	0.000 **
ACE language	1. ADEM-V	0.777	0.603	0.591	2.75	0.603	50.216	0.000	50.216	0.000 **
	2. 1 + Stereopsis	0.825	0.681	0.661	2.51	0.077	7.729	0.009	34.092	0.000 **
ACE visuospatial	1. ADEM-H	0.662	0.438	0.421	2.16	0.438	25.721	0.000	25.721	0.000 **

** *p* < 0.01.

**Table 5 jcm-11-02624-t005:** ROC curve analysis about DEM, stereopsis, and OCT.

Variable	AUC	SE	*p*	95%CI		Sensitivity	Specificity	Cut-Off Point
				Lower	Upper			
ADEM_V	0.914	0.047	0.000 **	0.823	1.00	0.857	0.867	35.5
ADEM_H	0.924	0.047	0.000 **	0.831	1.00	0.905	0.933	38.5
ADEM_Ratio	0.902	0.049	0.000 **	0.806	0.997	0.762	0.933	1.21
OCT_O_I	0.795	0.079	0.009 **	0.604	0.913	1	0.429	243
Stereopsis	0.794	0.084	0.012 *	0.585	0.913	0.857	0.733	100

* *p* < 0.05, ** *p* < 0.01.

## Data Availability

The datasets used during the current study are available from the corresponding author.

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
