# Peer review of "Correlating Ocular Physiology and Visual Function with Mild Cognitive Loss in Senior Citizens in Taiwan"

_jcm, 2022, doi:10.3390/jcm11092624_

Round 1

Reviewer 1 Report

The association of VI with cognitive decline has been interesting research topic due to aging society.  However, there are many researches conducted previously to see the association between VI and cognitive decline. In addition, examing 36 participants in the study may not be enough considering previous studies. Also suggestion that ADEM or stereopsis test can be diagnostic tool for identifying patients with mild cognitive impairment may not be persusive as no data of refractive error of the participants. . I here would like to provide a few more comments.

Overall Comments

The association of VI with cognitive decline has been interesting research topic due to aging society.  However, there are many researches conducted previously to see the association between VI and cognitive decline. In addition, examing 36 participants in the study may not be enough considering previous studies. Also suggestion that ADEM or stereopsis test can be diagnostic tool for identifying patients with mild cognitive impairment may not be persusive as no data of refractive error of the participants. . I here would like to provide a few more comments.

Introduction

Rationale of this study was needs more throughtly be written as there are many previous studies in this area. Differentiation from other study is not clear, therefore significance of this study is weak. 

Method

  1. Considering the age of the participants, there will be variables such as any cataract surgery which may affect accommodative ability. However there is no comment for those variables.
  2. Status of refractive error may be one of the important factors in visual fuction, without controlling refraction, the result could be biased. Therefore refractive errors should be equally distributed between groups in the study.

Results

  1. No demographic data such as participants' refractive error, and binocular vision test result which can all affect to the ADEM test result.
  2. Figure 2 and 3 needs to be redrawn.

Discussion

  1. The sentence of “outer and inferior macular layer as having excellent discrimination ability in predicting cognitive state under the linear regression and ROC curve” is repeat of result, rather than discussion. Also this is just observation, not explaning why it happen.
  2. The sentence of “ADEM, stereopsis, and macular thinness interact to some extent”. Stereopis could be binocular vision problem, therefore ADEM can be affected which is well known fact as commneted. However this finding also needs more explanation rather than stating the findings.

Author Response

Thank you for very constructive suggestions. Most of them have been revised according to the comments of the reviewers

Reviewer Comments

response

1

The association of VI with cognitive decline has been interesting research topic due to aging society. However, there are many researchers conducted previously to see the association between VI and cognitive decline.

Thank you for very constructive suggestions. Most of them have been revised according to the comments of the reviewers
However, this study explored the relationship between visual function and the degradation of ocular physiology and cognitive function, but did not explore the relationship between visual impairment and cognitive function degradation

2

examing 36 participants in the study may not be enough considering previous studies.

Revised

2.3. Data Analysis and Statistical Analysis

G*Power analysis was used to determine the sample size in this study, the condition was set at the effect size d=0.8, α=0.05, power(1-β)=0.95.

Results

G*Power analysis showed that there was still enough power after recalculation and adjustment.

ADEM or stereopsis test can be diagnostic tool for identifying patients with mild cognitive impairment may not be persusive as no data of refractive error of the participants.

Revised

In the Research subjects paragraph, we set some criteria for inclusion, 

All participants had binocular and monocular distance and near visual acuity of 0.8 or better, spherical power ranged from -5.00D to +2.00D, and astigmatism < 1.00D.

3.1. Ocular Physicology between ACE-abnormal and ACE-normal groups

All data were collected after confirming that subjects had no existing eye diseases and risks; at the same time, all participants had eligible refractive errors and visual acuity.

1

Introduction

Rationale of this study was needs more throughtly be written as there are many previous studies in this area. Differentiation from other study is not clear, therefore significance of this study is weak.

This study explored the relationship between visual function and the degradation of ocular physiology and cognitive function, but did not explore the relationship between visual impairment and cognitive function degradation

2

Method

1.     Considering the age of the participants, there will be variables such as any cataract surgery which may affect accommodative ability. However there is no comment for those variables.

2.     Status of refractive error may be one of the important factors in visual fuction, without controlling refraction, the result could be biased. Therefore refractive errors should be equally distributed between groups in the study.

Revised

In the Research subjects paragraph, we set some criteria for inclusion, 

All participants had binocular and monocular distance and near visual acuity of 0.8 or better, spherical power ranged from -5.00D to +2.00D, and astigmatism < 1.00D.

3.1. Ocular Physicology between ACE-abnormal and ACE-normal groups

All data were collected after confirming that subjects had no existing eye diseases and risks; at the same time, all participants had eligible refractive errors and visual acuity.

3

Results

1.No demographic data such as participants' refractive error, and binocular vision test result which can all affect to the ADEM test result.

2.Figure 2 and 3 needs to be redrawn.

Revised

In the Research subjects paragraph, we set some criteria for inclusion, 

All participants had binocular and monocular distance and near visual acuity of 0.8 or better, spherical power ranged from -5.00D to +2.00D, and astigmatism < 1.00D.

3.1. Ocular Physicology between ACE-abnormal and ACE-normal groups

All data were collected after confirming that subjects had no existing eye diseases and risks; at the same time, all participants had eligible refractive errors and visual acuity.

Revised

4

Discussion

1.The sentence of “outer and inferior macular layer as having excellent discrimination ability in predicting cognitive state under the linear regression and ROC curve” is repeat of result, rather than discussion. Also this is just observation, not explaning why it happen.

2.The sentence of “ADEM, stereopsis, and macular thinness interact to some extent”. Stereopis could be binocular vision problem, therefore ADEM can be affected which is well known fact as commneted. However this finding also needs more explanation rather than stating the findings.

We should point out that procedures for monitoring cognitive impairment through readily available testing of eye movement and stereopsis, and OCT measurement of retinal thickness, must be done as a whole. In a suitably well-equipped eye clinic, this is not a problem. Otherwise, the correlation between single tests and cognitive function will not be as high as desired (see Table 5, the correlation coefficient is between 0.40 and 0.47), and referrals for additional tests may become necessary.

Reviewer 2 Report

This is a novel study for eliciting MCI and deserves a wider audience. It is well articulated, flows well, and has significant statistical analysis using multiple known tools, giving a solid impression of significant findings in abnormal visual capacity in seniors' eyesight. The beauty of this paper is that the "ADEM tool can effectively identify cognitive impairment, is inexpensive, rapid, and simple."

Please check the classifications in the Abstract - wrt Results. The numeral (3) appears duplicated when it should be (2). Novel paper with significant finding using the ADEM tool for screening of MCI.

Author Response

Thank you for very constructive suggestions. Most of them have been revised according to the comments of the reviewers.

Please check the classifications in the Abstract - wrt Results. The numeral (3) appears duplicated when it should be (2). Novel paper with significant finding using the ADEM tool for screening of MCI.

Revised

Reviewer 3 Report

Thank you for the opportunity of reviewing this manuscript . I would like to appreciate the authors for this macroscopic study on a very important issue. In my opinion this is a piece of research of academic and clinical interest for visual function and cognitive loss in senior patients

In OCT examination, outer retina showed more significantly different and  center fovea was not significantly different. please more discuss this results

In OCT examination, the optic nerve head analysis was not performed?

Author Response

Thank you for very constructive suggestions. Most of them have been revised according to the comments of the reviewers.

Question 1

In OCT examination, outer retina showed more significantly different and  center fovea was not significantly different. please more discuss this results

Page 9 Discussion~ Revised

According the the previous studies and the results in this study, we can assume that the thinning of the ganglion cell layer leads to the disappearance of neurons, hereby affecting the dorsal and the ventral pathway; in addition,  these neural changes contribute to age-related losses of low-level visual functions and even higher-order visual perceptions,including face perception[37], motion processing[38-40], and reading speed[41].

Question 2

In OCT examination, the optic nerve head analysis was not performed?

Those with an optic disc ratio not within the normal range may be a high-risk group with high myopia or glaucoma, were also excluded and referred.

Round 2

Reviewer 3 Report

All my previous concerns have been addressed. Good work, well done!